# Genetic Variability and Recombination of the *NSP2* Gene of PRRSV-2 Strains in China from 1996 to 2021

**DOI:** 10.3390/vetsci10050325

**Published:** 2023-04-29

**Authors:** Hang Zhang, Qin Luo, Yajie Zheng, Huiyang Sha, Gan Li, Weili Kong, Liangzong Huang, Mengmeng Zhao

**Affiliations:** 1School of Life Science and Engineering, Foshan University, Foshan 528000, China; hangzh2022@163.com (H.Z.); luoqin121104@163.com (Q.L.); zhengyajie2022@163.com (Y.Z.); huiyangsha2022@163.com (H.S.); ligan1227@163.com (G.L.); 2Gladstone Institutes of Virology and Immunology, University of California, San Francisco, CA 94158, USA; weili.kong@gladstone.ucsf.edu

**Keywords:** porcine reproductive and respiratory syndrome virus, *NSP2* gene, genetic variability, recombination, phylogeny

## Abstract

**Simple Summary:**

Genetic variability and recombination of the *NSP2* gene are of great significance in gaining an in-depth understanding of the prevalence of PRRSV in China over the past 25 years. We compared the nucleotide and amino acid homologies of the NSP2 sequences of different PRRSV-2 lineages, and examined phylogenetic relationships based on an analysis of the NSP2 sequences of 122 strains. What is more, recombination analysis revealed the occurrence of five recombinant events among the 135 selected PRRSV-2 strains. These results provide a theoretical foundation for evolution and epidemiology of the spread of PRRSV.

**Abstract:**

Porcine reproductive and respiratory syndrome (PRRS) is one of the most serious infectious diseases that detrimentally affects the pig industry worldwide. The disease, which is typically difficult to control, is an immunosuppressive disease caused by the porcine reproductive and respiratory syndrome virus (PRRSV), the genome of which (notably the *NSP2* gene) undergoes rapid mutation. In this study, we sought to determine the genetic variation in the PRRSV-2 *NSP2* gene in China from 1996 to 2021. Strain information was obtained from the GenBank database and analyzed from a molecular epidemiological perspective. We compared the nucleotide and amino acid homologies of the NSP2 sequences of different PRRSV-2 lineages, and examined phylogenetic relationships based on an analysis of the NSP2 sequences of 122 strains. The results revealed that NADC-30-like strains, which are represented by lineage 1, and HP-PRRSV strains, which are represented by lineage 8, were the most prevalent in China from 1996 to 2021. Close similarities were detected in the genetic evolution of lineages 3, 5, and 8. For nucleotide and amino acid sequence comparisons, we selected representative strains from each lineage, and for the NSP2 among different PRRSV-2 strains, we accordingly detected homologies of 72.5–99.8% and 63.9–99.4% at the nucleotide and amino acid levels, respectively, thereby indicating certain differences in the degrees of NSP2 amino acid and nucleotide variation. Based on amino acid sequence comparisons, we identified deletions, insertions, and substitutions at multiple sites among the NSP2 sequences of PRRSV-2 strains. Recombination analysis revealed the occurrence of five recombinant events among the 135 selected PRRSV-2 strains, and that there is a high probability of recombination of lineage 1 strains. The findings of this study enabled us to gain an in-depth understanding of the prevalence of PRRSV in China over the past 25 years and will contribute to providing a theoretical basis for evolution and epidemiology of the spread of PRRSV.

## 1. Introduction

Porcine reproductive and respiratory syndrome (PRRS) is a highly prevalent infectious disease caused by the porcine reproductive and respiratory syndrome virus (PRRSV). PRRSV infection causes immunosuppression; reproductive disorders in pregnant sows, including miscarriage, premature birth, and mummified fetuses; and respiratory diseases in piglets, thereby having particularly detrimental impacts on the pig industry worldwide. PRRS was first discovered in North Carolina in 1987, and the LV and VR2332 strains were subsequently isolated from infected pigs in Europe and America in 1991 and 1992, respectively [1,2]. At present, PRRSV-1 and PRRSV-2 were classified as the species Betaarterivirus suid 1 and Betaarterivirus suid 2, respectively. PRRSV-2 strains are mainly prevalent in China, wherein PRRSV was isolated for the first time in 1996 [3]. In 2006, a highly pathogenic PRRSV (HP-PRRSV) was detected, which was associated with a high fatality rate among piglets [4,5]. More recently, in 2012, a new strain type, NADC30, appeared in China [6]. PRRSV continues to recombine and mutate, which accordingly presents considerable challenges with respect to the prevention and control of PRRS [7].

PRRSV, a single-stranded positive-sense RNA virus, is a member of the genus *Arterivirus* in the family *Arterivirus* of the order *Nidoviridae*. It comprises a 15 kb genome containing 11 open reading frames (ORFs), namely, ORF1a, ORF1b, ORF2a, ORF2b, ORF3-7, ORF5a, and ORF1aTF, that overlaps the non-structural protein (NSP) 2-encoding region of ORF1a [8]. Among these, ORF1a and ORF1aTF translationally encode at least 12 NSPs, including NSP1α, NSP1β, NSP2-related proteins (NSP2N, NSP2TF, and NSP2), NSP3, NSP4, NSP5, NSP6, NSP7α, NSP7β, and NSP8 [9,10]; NSP2 at 2.9 kb in length is the longest. NSP2 and ORF5 are highly variable and ORF5 is associated with the neutralizing epitope. They are usually used as target genes for PRRSV molecular epidemiological surveillance [11]. Since the time of its isolation in the United States in 2001, the MN184 strain has been found to have discontinuously lost 131 amino acids in NSP2 sequence [12], whereas isolates obtained during the HP-PRRSV outbreak in China in 2006 were found to have discontinuously lost 30 amino acids from NSP2 sequence [5]. Similarly, the NSP2 sequence of the NADC34-like PRRSV strains recently isolated in China was found to have undergone a continuous loss of 100 amino acids. Studies have demonstrated that the recombinant strain HLJ/2017/1127a is a result of the recombination of the FZ06A and QYYZ strains. The recombination breakpoint is located at 1892–2730 (1742–2442) [13].

Shi et al. proposed a systematic classification of PRRSV-2 based on *ORF5* gene in 2011, dividing PRRSV-2 into 9 lineages and 37 subfamilies [14]. Given the increasing number of PRRSV recombinant strains and increases in recombinant frequency in recent years, the epidemic situation of PRRSV in our country has become progressively more complex, indicating that recombination of PRRSV has played an important role in virus evolution [15,16]. Moreover, this continual variation and recombination of the PRRSV genome also hampers measures currently taken to control the incidence and spread of PRRS. Consequently, timely monitoring of the patterns of PRRSV mutation is essential for ensuring effective epidemic evolution and epidemiology. To this end, in this study, we compared the nucleotide and amino acids homologies of PRRSV-2 NSP2 among the strains of different lineages, and analyzed phylogenetic relationships, thereby enabling us to gain more understanding of genetic variation in the NSP2 protein. These findings will contribute to establishing a theoretical basis for assessing future epidemic trends in PRRS and for identifying changes in the NSP2 sequence that these viruses use to evade host immunity, thereby perpetuating the genetic evolution of PRRSV.

## 2. Materials and Methods

### 2.1. The Dataset 

One hundred and twenty-two Chinese PRRSV-2 strains and 13 US PRRSV-2 strains in the NCBI website GenBank database were chosen, including those of lineage 1, 3, 5, and 8 (Table 1 and Table 2). There are a few reasons why these strains were selected. First, the year needs to be included in different years from 1996 to the present year. Second, the strains are representative of PRRSV strains frequently reported in other pieces of literature. Third, the commonly used vaccine strains needed to be selected.

### 2.2. Phylogenetic Analysis

Phylogenetic analysis of the *NSP2* gene was based on the reference strains sequence information shown in Table 1. The comparison was first performed using the Clustal W method in the MegAlign function of DNAStar software (version 7.0) and then using the neighbor-joining (NJ) method of MEGA software (version 7.0) with 1000 bootstrap replicates. It was then analyzed using the Maximum Likelihood (ML) method of PhyloSuite software (version 1.2.2) with 1000 bootstrap replicates.

### 2.3. Alignment of NSP2 Nucleotide Sequences

For the purpose of determining similarities among the NSP2 nucleotide sequences of different PRRSV lineages, we analyzed the reference strains information shown in Table 1 based on the Clustal W method in the MegAlign function of DNAStar software.

### 2.4. Alignment of NSP2 Amino Acid Sequences

Similarities among NSP2 amino acid sequences were analyzed using the Clustal W method, based on reference strains information, and multi-sequence alignment analysis was performed using the BioEdit software (version 7.2).

### 2.5. Recombination Analysis

When potential recombinant events were detected based on RDP software (version 4.0), GENECONV, BootScan, MaxChi, Chimera, SiScan, and 3eq analyses, five or more methods were identified as genetic recombination and *P <* 0.05 in RDP software. The strains thus identified were considered recombinant strains. In addition, we used SimPlot (version 3.5.1) to confirm the detected recombination events.

## 3. Results

### 3.1. Phylogenetic Analysis

Based on the global PRRSV classification system and NSP2 sequence information in the GenBank database, we selected the NSP2 sequences of 122 PRRSV-2 strains for phylogenetic analysis (Table 1). The phylogenetic tree constructed using these sequences revealed that the PRRSV-2 strains prevalent in China could be classified into four lineages, namely, lineages 1, 3, 5, and 8 (Figure 1 and Figure 2). Among these, lineage 3 is represented by GM2-2011, QYYZ-2011, and FJFS-2012, which appear to be closely related, whereas lineage 1 and lineage 8 strains appear to be separated by comparatively large genetic distances. Since the advent of PRRSV in China, lineage 8, which includes the classical PRRSV strains (CH-1a-like) prevalent before 2006 and the HP-PRRSV-like strains circulating after 2006, has predominated. However, since 2013, lineage 1 strains, also referred to as NADC30-like strains, have spread rapidly across the country, with clinical detection rates comparable to those of HP-PRRSV-like lineage 8 strains. 

### 3.2. Nucleotide Similarity 

In order to further examine the genetic variation in NSP2 that has occurred during the course of PRRSV evolution, we selected 15 strains from each of the aforementioned four lineages for nucleotide homology analyses, and thereby determined evolutionary relationships among the different lineages at the nucleotide level (Figure 3). We accordingly detected a nucleotide homology of between 72.5% and 99.8% among the NSP2 protein of different PRRSV-2 strains, of which the CHsx1401-2014 strain showed the lowest homology of 72.5% with the QYYZ-2011 and GM2-2011 strains. Contrastingly, the nucleotide sequence of the BJ-4-1996 strain was found to be highly similar to that of the RespPRRS MLV-1994 strain, with a homology of 99.8%. With respect to each of the four assessed lineages, we obtained homology values of 86.0% to 91.2% among lineage 1 strains, 92.4% to 98.8% among lineage 3 strains, 99.4% to 99.8% among lineage 5 strains, and 92.3% to 99.6% among lineage 8 strains, of which the homologies of classic-type CH-1a-like strains were between 92.3% and 99.3% and those of HP-PRRSV-like strains were between 99.4% and 99.6%. The largest differences in NSP2 sequences were detected in lineage 1 (NADC-30-like) strains, and we speculate that this reflects extensive mutation and recombination among these strains. Lineages 3 and 5 were found to be highly similar at the nucleotide level, and the clinical detection of these two lineages in China is generally very low. 

### 3.3. Amino Acid Sequence Similarity

Similar to our analysis of nucleotide similarities, we also examined the genetic variation of the PRRSV NSP2 protein at the amino acid level among 15 representative strains of the four lineages, in order to gain an understanding of the evolutionary relationship among amino acids sequences in each lineage (Figure 4). The results revealed amino acid homologies of between 63.9% and 99.4% among the different PRRSV-2 strains, of which that between CHsx1401-2014 and GM2-2011 at 63.9% was the lowest. The highest similarities were detected between the TJ-2006 strain and strains HUN4-2006 and JXA1-2006, with homologies reaching 99.4%, which was found to be inconsistent with the differences among these strains detected at the nucleotide level. In this regard, it is conceivable that amino acid substitution between lineage 1 and 5 strains is more common. In terms of each lineage, we detected amino acid homologies of 80.8% to 87.8%, 88.8% to 98.3%, 98.7% to 99.3%, and 87.9% to 99.4% among lineage 1, 3, 5, and 8 strains, respectively. Of these, homologies of between 89.6% and 98.7%, and 99.2% and 99.4% were obtained for the classical type CH-1a-like and HP-PRRSV-like strains, respectively. The largest differences in NSP2 sequences were identified in lineage 1 (NADC-30-like) strains, which is consistent with the nucleotide alignment results.

### 3.4. Amino Acid Sequence Alignment

The NSP2 nucleotide sequences of the 15 selected strains of each lineage were initially translated into amino acid sequences, for which we subsequently performed a multi-sequence comparative analysis (Figure 5). We accordingly detected differences in the lengths of PRRSV-2 NSP2 amino acid sequences, characterized by significant variability in the sequences encoding NSP2, including deletions, insertions, and substitutions at different amino acid sites. As previously stated, lineage 1 and 8 NADC-30-like and HP-PRRSV-like strains are currently the most prevalent in China, and have long been detected in the country, during which time they have caused substantial economic losses in the pig industry. Based on our brief analysis of amino acid mutations in the highly pathogenic lineage 8 strains, we detected 30 discontinuous deletions at amino acid residues 481, and 533 to 561 in the HP-PRRSV representative strains HUN4-2006, JXA1-2006, and TJ-2006, which is consistent with the findings of a previous study [4]. Moreover, we identified high amino acid homologies of up to 99.4% among these strains, with up to only 10 amino acid differences. In the TJ-2006 strain, these changes are located at positions 8, 194, 213, 253, 296, 488, 504, 544, 745, and 796, at which we detected proline, asparagine, histidine, glutamic acid, leucine, valine, serine, alanine, and asparagine residues, respectively. In JXA1-2006, changes were identified at positions 8, 253, 296, 488, 745, and 796, at which we detected the substitution of threonine, valine, phenylalanine, valine, and glycine, respectively, whereas in the HUN4-2006 strain, substitutions of threonine, tyrosine, phenylalanine, methionine, and proline were detected at positions 8, 194, 213, 488, 504, and 544, respectively. Furthermore, in representative strains of lineage 1, we detected different degrees of amino acid deletion at positions 481, 320–323, 325–345, 347–381, 383–429, 431–434, and 502–520, whereas in representative lineage 3 strains, in addition to the deletion mutations, two amino acid deletions were found at positions 300 and 301, and there has been a continuous insertion of 36 amino acids at positions 817–852.

### 3.5. Recombinant Analysis

To gain a more complete picture of the recombination of PRRSV-2, we added some classic strains from the United States (Table 2). Recombinant analysis of the *NSP2* gene revealed seven potential recombinant events by RDP4 (Table 3), among which, five potential recombinant events was verified by SimPlot. According to the RDP4 results, the credibility of all seven recombination events is high with a statistical significance of *P <* 0.05. Events 2 and 3 were detected in lineage 1 strains, involving a recombination of lineage 1 and 8 sequences (Figure 6). The primary and secondary parental strains of recombinant strain SD-YL1712-2017 are BJ2021-2021 and JSTZ1712-12-2017, respectively; the primary and secondary parental strains of recombinant strain MN184A-2001 are NADC30-2008 and QHD3-2017, respectively. Similarly, recombinant events 4, 5, and 7 were detected in lineage 1 strains, resulting from recombination among these strains. The primary and secondary parental strains of recombinant strain NADC30-2017 are SD-A19-2015 and FJ1402-2014, respectively; the primary and secondary parental strains of recombinant strain SCya18-2018 are SD17-36-2017 and ZJXS1412-2014, respectively; and the primary and secondary parental strains of recombinant strain SCcd2020-2020 are JS2020-2020 and SCcd17-2017, respectively. Among lineage 8 strains, we detected recombinant events 1 and 6, resulting from recombination between lineage 5 and 8 strains. In the case of recombinant strain CH-1R-2008, the primary and secondary parental strains are GD3-2005 and CH-1a-1996, respectively, whereas the primary and secondary parental strains of recombinant strain Ingelvac-ATP-1999 are CC-1-2006 and HB-2(sh)-2001, respectively. However, SimPlot verified that no recombination occurred in events 3 and 5 (Figure 7).

## 4. Discussion

Our analysis of PRRSV evolution covering a period of approximately 25 years revealed apparent changes in the highly variable region of the NSP2 protein. In order to gain a comprehensive insight into evolutionary changes in the NSP2 genetic evolution, we performed phylogenetic analyses based on the NSP2 sequences of 122 selected PRRSV-2 strains, the results of which revealed the genetically close evolutionary distances of lineage 3 and 5 strains. Lineage 3 comprises variants that have emerged since 2010, the transmission of which has been recorded primarily in southern China (Jiangxi, Fujian, Guangdong, and Guangxi provinces) with a clinical detection rate of less than 10% [17,18]. In contrast, although lineage 5 (BJ-4-like/VR2332-like) strains appeared as early as 1996, these do not appear to be prevalent in China and have a low clinical detection rate [19]. In China, lineage 8 (HP-PRRSV-like) and lineage 1 (NADC30-like) strains have become the predominant strains, which is consistent with previous reports, which may be attributable to the fact that these strains are characterized by high genetic variation and recombinant properties, which are features facilitating the evasion of immune surveillance promoted by existing vaccines. Zhao et al. [20] reported that NADC30-like PRRSV has gradually become the most prevalent genotype in Sichuan Province. Yu et al. [21] proofed NADC30-like PRRSVs are undergoing a decrease in population genetic diversity. Zhou et al. [22] found that the NADC30 and the HP-PRRSV strains mainly circulated in southwest China.

For each of the four assessed lineages, we selected different representative strains for comparisons of nucleotide and amino acid sequences, which revealed nucleotide and amino acid homologies of 72.5% to 99.8% and 63.9% to 99.4%, respectively, among the NSP2 proteins of different PRRSV-2 strains. We speculate that these strains have undergone relatively limited mutation during the course of genetic evolution, and accordingly have been unable to effectively evade vaccine-mediated immunity and host immune surveillance, although further studies are necessary to ascertain specific details in this regard. Currently, lineage 1 and 8 strains are the prevalent epidemic strains in China, among which, the NADC30-like strains appear to be characterized by a high frequency to recombinant events, thereby enabling these viruses to adapt to changing environmental pressures during long-term evolution [6]. Further, it is speculated that the NSP2 sequence has also undergone a corresponding series of changes as a consequence of mutation and recombination of the different lineage 1 strains. Moreover, we detected high amino acid homologies between lineages 3 and 5 strains, which is consistent with the patterns observed in the nucleotide comparison.

Compared with the classical PRRSV strains, HP-PRRSV strains are characterized by a discontinuous deletion of (29 + 1) amino acids, whereas for NADC30-like strains there is evidence of a discontinuous deletion of 131 amino acids. Lineage 3 strains of PRRSV-2, specifically GM2-2011, QYYZ-2011, and FJFS-2012, exhibit 37 amino acid insertions at positions 817–853 of NSP2. Comparison of amino acid sequences among different PRRSV-2 strains reveals variations in deletions, insertions, and substitutions at multiple sites, which suggest differences in antigenicity. 

There are some reports showing the recombination of NSP2 [13,23]. Taking NADC30-like strains as an example, new deletion of NADC30-like strains occurs, which makes it easier to recombine. Recombinant analysis performed for PRRSV-2 NSP2 revealed a total of five recombinant events among the 135 selected strains, a majority of which appear to have occurred between lineage 1 and 8 strains, possibly because there are many types of PRRSV vaccines in the Chinese market (inactivated vaccines: KV; activated vaccines: Ingelvac PRRS MLV, MLV CH-1R, MLV R98, MLV JXA1-R, MLV TJM-F92, MLV HuN4-F112, MLV GDr180), and the abuse of vaccines is also the main reason why PRRSV is prone to recombination. There is no solid evidence that the changes in *NSP2* gene affects the recombination of PRRSV-2. Maybe through reverse genetics, mutation, or deletion of NSP2, the results of recombination can be further observed. So far, no such result has been reported. We will conduct such research in the future. Fang et al. found a high frequency of interlineage recombination hot spots in NSP9, so we hypothesize that changes in the *NSP2* gene can affect PRRSV recombination. For the clinical severity, variation of NSP2 sequence was not related to viral virulence. Studies have shown that virulence is related to PRRSV NSP9 and NSP10 (NSP9 and NSP10 contribute to the fatal virulence of HP-PRRSV emerging in China), and two residues in NSP9 contribute to the enhanced replication and pathogenicity of HP-PRRSV [24]. Although HP-PRRSV derived vaccines have contributed to reducing the severity of clinical signs and restricting the spread of PRRS to a certain extent, their efficacy still falls far short of that initially anticipated. Moreover, the frequent use of vaccines can have a number of undesirable side effects, including the anti-virulence of live attenuated vaccines [25,26], viral recombination, and a significant increase in the rate of mutation [27,28,29].

Compared with the results of Jiang [13], our study focuses on the PRRSV-2 *NSP2* gene, which we have divided into four lineages. We conducted nucleotide and amino acid similarity analyses, as well as sequence alignment and recombination analysis on NSP2. Although today we have a better understanding of the clinical pathogenesis of PRRS, molecular epidemiology, viral proliferation, pathogenesis, immune response, and immune evasive mechanisms, PRRSV is still not effectively controlled. In the past, investigators tended to focus on amino acid mutations in the hypervariable region (NSP2, ORF5) of the virus genome; however, in recent years, several strain recombination events have been discovered, and the frequency of recombination appears to be rising. In this regard, Zhao et al. [30] have confirmed that the highly pathogenic JL580 strain has a mixed genetic background of NADC30-like, HP-PRRSV, and local strains, and Chen et al. [31] concluded that HBap4-2018 is a new PRRSV strain with HP-PRRSV and NADC30-like PRRSV as primary and secondary parental strains, respectively, which retains most of the virulence-related regions of the HP-PRRSV genome and has been shown to be highly pathogenic to piglets. Moreover, Li et al. [32] have established that the HNyc15 strain is derived from the recombination of VR2332 and CH-1a gene fragments. The increases in recombinant strains and high rates of recombination in recent years indicate that recombination has played a very important role in the evolution of the virus. PRRSV *NSP2* gene is prone to deletion, insertion, and recombination, and recombination plays an important role in the evolution of viruses. Therefore, we speculate that the change in *NSP2* gene may affect the evolution of PRRSV. From the perspective of the prevention and control of PRRSV, it is accordingly imperative to ascertain the epidemiological characteristics of PRRSV, establish the molecular mechanisms underlying the high variability and recombinant frequency, and elucidate determinants of the variable virulence observed among different PRRSV strains.

## 5. Conclusions

From 1996 to 2021, PRRSV-2 isolates in China have been categorized into lineages 1, 3, 5, and 8, based on variations in their *NSP2* genes. Lineages 1 and 8 have been found to be predominant in the prevalence of PRRS in China. Additionally, the removal of the NSP2 sequence prior to 2006 has been identified as a key factor driving the evolution of PRRSV and has also contributed to the evolution of virus. In recent years, recombination has played a significant role in the evolution of PRRSV-2. Therefore, monitoring the *NSP2* gene of PRRSV-2 can serve as a reference for effectively controlling PRRS in China in the future.

## Figures and Tables

**Figure 1 vetsci-10-00325-f001:**
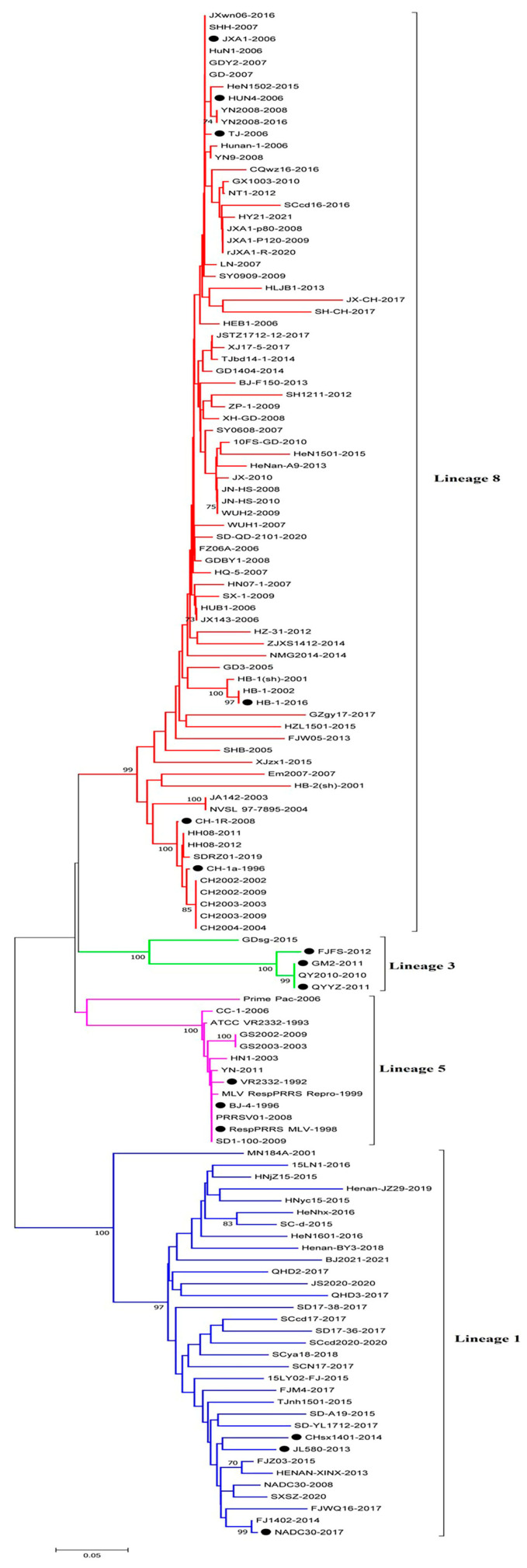
A NJ phylogenetic tree was constructed based on the NSP2 sequences of 122 PRRSV strains, using MEGA software (version 7.0) with 1000 bootstrap replicates. The circles (●) refer to the representative virus strains of each lineage in China.

**Figure 2 vetsci-10-00325-f002:**
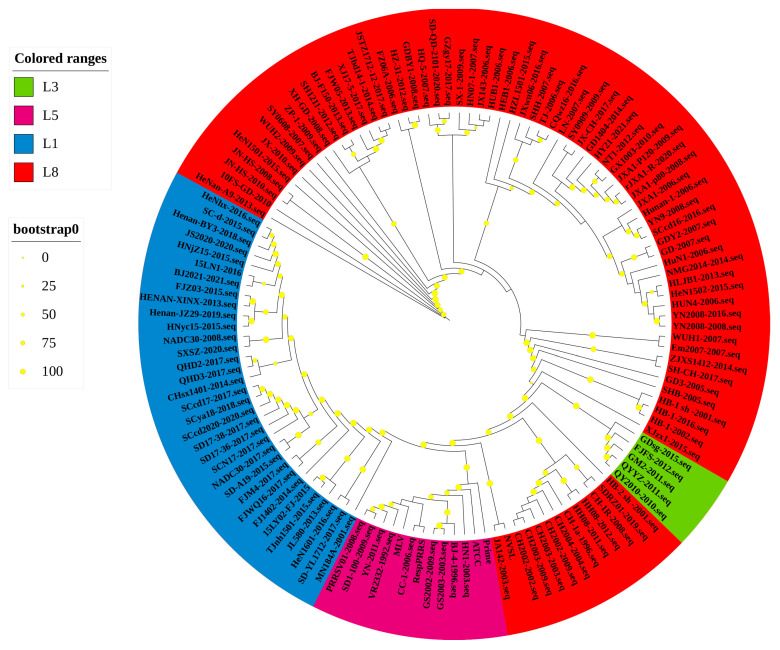
A ML phylogenetic tree was constructed based on the NSP2 sequences of 122 PRRSV strains, using PhyloSuite software (version 1.2.2) with 1000 bootstrap replicates.

**Figure 3 vetsci-10-00325-f003:**
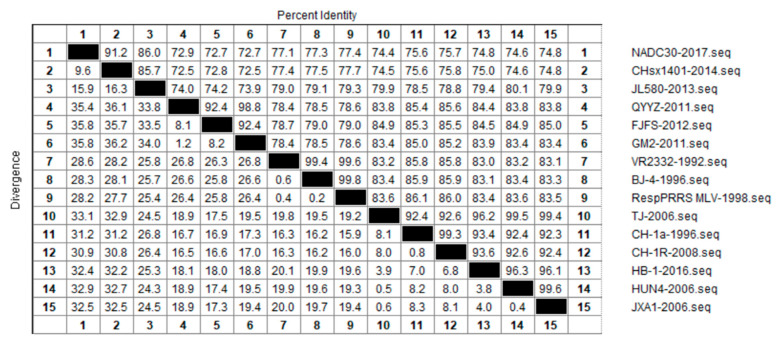
Results of similarity analysis performed for NSP2 nucleotide sequences from representative strains of PRRSV lineages 1, 3, 5, and 8. Clustal W method in MegAlign function of DNAStar software was used to analyze the similarity of NSP2 nucleotides. The 15 representative strains selected were CHsx1401-2014, JL580-2013, NADC30-2008 of lineage 1, GM2-2011, QYYZ-2011, FJFS-2012 of lineage 3, VR2332-1992, BJ-4-1996, RespPRRS MLV-1998 of lineage 5. CH-1a-1996, CH-1R-2008, CH2002-2009 of classical CH-1a-like in lineage 8, JXA1-2006, TJ-2006, HUN4-2006 of highly pathogenic HP-PRRSV-like in lineage 8.

**Figure 4 vetsci-10-00325-f004:**
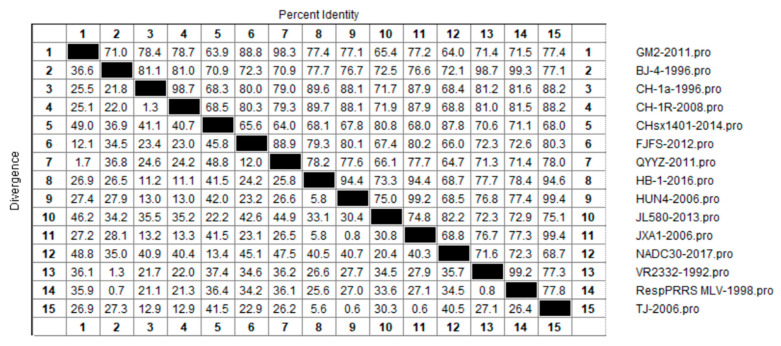
Results of similarity analysis performed for NSP2 amino acid sequences from representative strain of PRRSV lineages 1, 3, 5, and 8. Clustal W method in MegAlign function of DNAStar software was used to analyze the similarity of NSP2 amino acids. The 15 representative strains selected were CHsx1401-2014, JL580-2013, NADC30-2008 of lineage 1, GM2-2011, QYYZ-2011, FJFS-2012 of lineage 3, VR2332-1992, BJ-4-1996, RespPRRS MLV-1998 of lineage 5. CH-1a-1996, CH-1R-2008, CH2002-2009 of classical CH-1a-like in lineage 8, JXA1-2006, TJ-2006, HUN4-2006 of highly pathogenic HP-PRRSV-like in lineage 8.

**Figure 5 vetsci-10-00325-f005:**
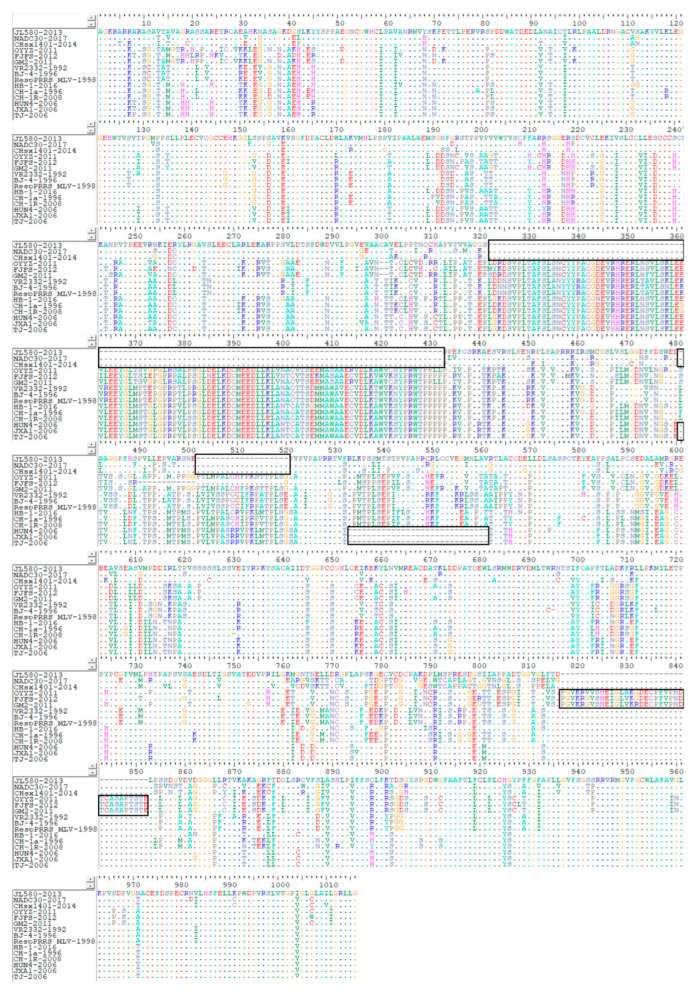
NSP2 amino acid sequence alignment for representative strains of PRRSV lineages 1, 3, 5, and 8. BioEdit software (version 7.2) was used to analyze amino acids of NSP2. The 15 representative strains selected were CHsx1401-2014, JL580-2013, NADC30-2008 of lineage 1, GM2-2011, QYYZ-2011, FJFS-2012 of lineage 3, VR2332-1992, BJ-4-1996, RespPRRS MLV-1998 of lineage 5. CH-1a-1996, CH-1R-2008, CH2002-2009 of classical CH-1a-like in lineage 8, JXA1-2006, TJ-2006, HUN4-2006 of highly pathogenic HP-PRRSV-like in lineage 8. The contents in the black box represent the deletion and insertion of amino acids.

**Figure 6 vetsci-10-00325-f006:**
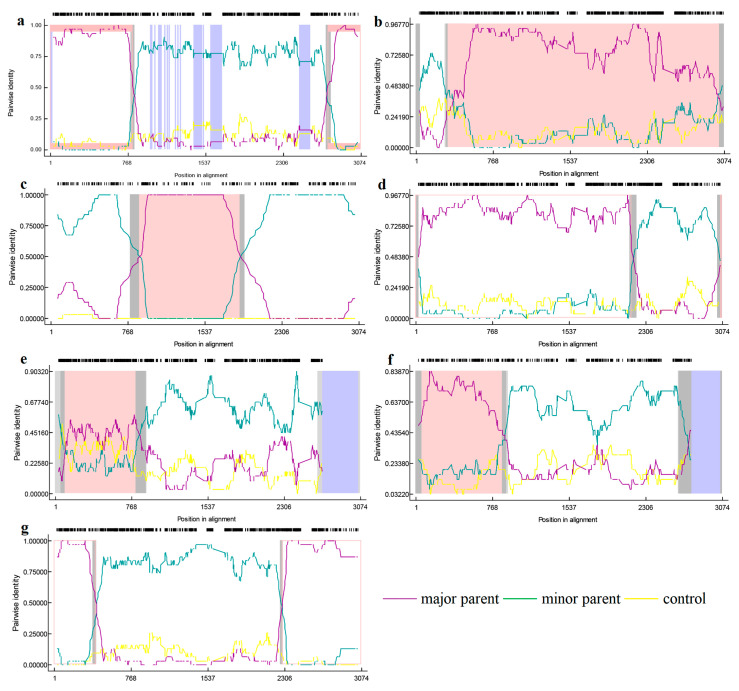
RDP (version 4.0) verification of *NSP2* gene recombination events. The horizontal axis represents position in alignment, the vertical axis represents pairwise identity. (**a**) is the RDP analysis result of the recombinant strain CH-1R; (**b**) is the RDP analysis result of the recombinant strain SD-YL1712-2017; (**c**) is the RDP analysis result of the recombinant strain MN-184A-2001; (**d**) is the RDP analysis result of the recombinant strain NADC30-2017; (**e**) is the RDP analysis result of the recombinant strain SCya18-2018; (**f**) is the RDP analysis result of the recombinant strain Ingelvac-ATP-1999; (**g**) is the RDP analysis results of the recombinant strain SCcd2020-2020.

**Figure 7 vetsci-10-00325-f007:**
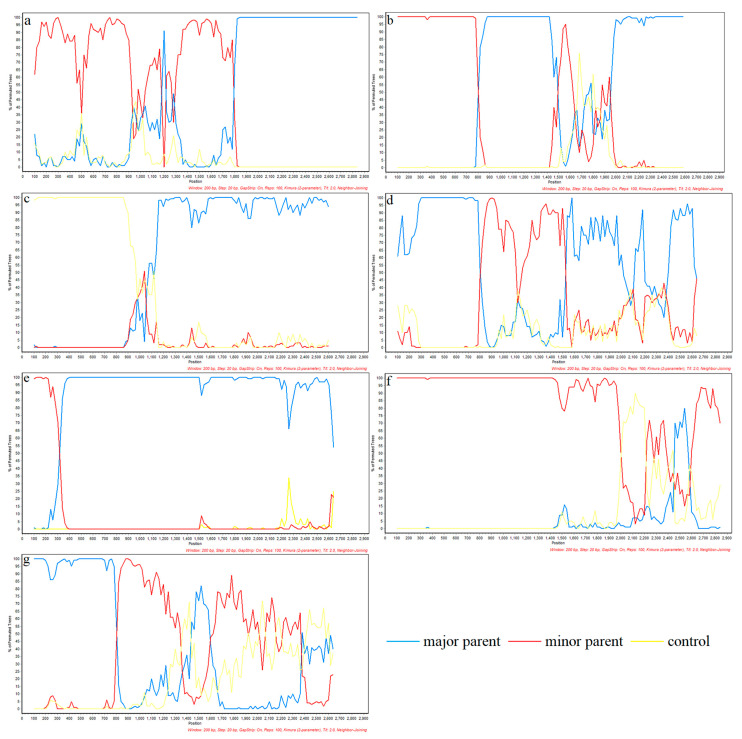
Simplot verification of *NSP2* gene recombination events. Simplot (version 3.5.1) was used to confirm the recombination events which are detected in RDP (version 4.0) software. The horizontal axis represents position, the vertical axis represents percentage of permuted trees. The different colors depict different PRRSV strains, and blue lines represent the major parent; red lines represent minor parent. (**a**) is the Simplot validation result of the recombinant strain CH-1R; (**b**) is the Simplot validation result of the recombinant strain SD-YL1712-2017; (**c**) is the Simplot validation result of the recombinant strain MN-184A-2001; (**d**) is the Simplot validation result of the recombinant strain NADC30-2017; (**e**) is the Simplot validation result of the recombinant strain SCya18-2018; (**f**) is the Simplot validation result of the recombinant strain Ingelvac-ATP-1999; (**g**) is the Simplot validation results of the recombinant strain SCcd2020-2020.

**Table 1 vetsci-10-00325-t001:** The reference sequence of each PRRSV NSP2 strains in China.

Year	Aera	Sequence	Strain	Genbank Accession Number	Based on the ORF5 Lineage
1996	China	NSP2	CH-1a	AY032626	8
1996	China	NSP2	BJ-4	AF331831	5
2001	China	NSP2	HB-1(sh)	AY150312	8
2001	China	NSP2	HB-2(sh)	AY262352	8
2002	China	NSP2	CH2002	EU880438.2	8
2002	China	NSP2	HB-1	HQ233605	8
2003	China	NSP2	HN1	AY457635.1	5
2003	China	NSP2	CH2003	EU880440.2	8
2003	China	NSP2	GS2003	EU880442	5
2004	China	NSP2	NVSL 97-7895	AY545985	8
2004	China	NSP2	CH2004	EU880439.2	8
2005	China	NSP2	GD3	GU269541	8
2005	China	NSP2	SHB	EU864232.1	8
2006	China	NSP2	Ingelvac ATP	DQ988080	8
2006	China	NSP2	Prime Pac	DQ779791	8
2006	China	NSP2	HEB1	EF112447	8
2006	China	NSP2	CC-1	EF153486	8
2006	China	NSP2	FZ06A	MF370557	8
2006	China	NSP2	JX143	EU708726	8
2006	China	NSP2	TJ	EU860248	8
2006	China	NSP2	HUB1	EF075945	8
2006	China	NSP2	Hunan-1	EU200965.1	8
2006	China	NSP2	JXA1	EF112445	8
2006	China	NSP2	HuN1	EF198112.1	8
2006	China	NSP2	HUN4	EF635006.1	8
2007	China	NSP2	Em2007	EU262603	8
2007	China	NSP2	SHH	EU106888	8
2007	China	NSP2	LN	EU109502	8
2007	China	NSP2	HN07-1	KX766378	8
2007	China	NSP2	SY0608	EU144079	8
2007	China	NSP2	GD	EU825724.1	8
2007	China	NSP2	HQ-5	EU255920.1	8
2007	China	NSP2	GDY2/2007	EU200964.1	8
2007	China	NSP2	WUH1	EU187484	8
2008	China	NSP2	XH-GD	EU624117	8
2008	China	NSP2	YN2008	EU880435	8
2008	China	NSP2	JN-HS	HM016158	8
2008	China	NSP2	YN9	GU232738	8
2008	China	NSP2	JXA1-p80	FJ548853	8
2008	China	NSP2	GDBY1	GQ374442	8
2008	China	NSP2	CH-1R	EU807840	8
2008	China	NSP2	PRRSV01	FJ175687	5
2009	China	NSP2	SY0909	HQ315837	8
2009	China	NSP2	ZP-1	HM016159	8
2009	China	NSP2	SX-1	GQ857656	8
2009	China	NSP2	WUH2	EU678352	8
2009	China	NSP2	CH2003	EU880440	8
2009	China	NSP2	JXA1-P120	KC422727	8
2009	China	NSP2	SD1-100	GQ914997	5
2009	China	NSP2	GS2002	EU880441	5
2009	China	NSP2	CH2002	EU880438	8
2010	China	NSP2	JN-HS	HM016158	8
2010	China	NSP2	10FS-GD	JX192634	8
2010	China	NSP2	JX	JX317649	8
2010	China	NSP2	QY2010	JQ743666	3
2010	China	NSP2	GX1003	JX912249	8
2011	China	NSP2	YN	JX857698	5
2011	China	NSP2	GM2	JN662424	3
2011	China	NSP2	QYYZ	JQ308798	3
2011	China	NSP2	HH08	JX679179	8
2012	China	NSP2	FJFS	KP998476	3
2012	China	NSP2	HZ-31	KC445138	5
2012	China	NSP2	SH1211	KF678434	8
2012	China	NSP2	NT1	KP179402	8
2013	China	NSP2	BJ-F150	KP890342	8
2013	China	NSP2	HeNan-A9	KJ546412	8
2013	China	NSP2	HENAN-XINX	KF611905	1
2013	China	NSP2	HLJB1	KT351740	8
2013	China	NSP2	JL580	KR706343.1	1
2013	China	NSP2	FJW05	KP860911	1
2014	China	NSP2	NMG2014	KM000066	8
2014	China	NSP2	GD1404	MF669720	8
2014	China	NSP2	ZJXS1412	MF669722	8
2014	China	NSP2	CHsx1401	KP861625	1
2014	China	NSP2	TJbd14-1	KP742986	8
2014	China	NSP2	FJ1402	KX169191	1
2015	China	NSP2	XJzx1	KX689233	8
2015	China	NSP2	HNyc15	KT945018	3
2015	China	NSP2	SC-d	MF375261	8
2015	China	NSP2	HeN1501	MF766472	8
2015	China	NSP2	HeN1502	MF766473	8
2015	China	NSP2	SD-A19	MF375260	8
2015	China	NSP2	HZL1501	MF669721	8
2015	China	NSP2	TJnh1501	KX510269	8
2015	China	NSP2	GDsg	KX621003	3
2015	China	NSP2	15LY02-FJ	KU215417	8
2015	China	NSP2	HNjZ15	KT945017	1
2015	China	NSP2	FJZ03	KP860909	1
2016	China	NSP2	YN2008	EU880435	8
2016	China	NSP2	HeNhx	KX766379	1
2016	China	NSP2	HeN1601	MF766474	8
2016	China	NSP2	CQwz16	KY620008.1	8
2016	China	NSP2	SCcd16	KY620011.1	8
2016	China	NSP2	HB-1	AH015834.2	8
2016	China	NSP2	JXwn06	EF641008	8
2016	China	NSP2	15LN1	KX815423	8
2017	China	NSP2	SD-YL1712	MT708500	8
2017	China	NSP2	SD17-36	MH121061	8
2017	China	NSP2	SD17-38	MH068878	8
2017	China	NSP2	QHD2	MH167387	8
2017	China	NSP2	JSTZ1712-12	MK906026	8
2017	China	NSP2	QHD3	MH167388	8
2017	China	NSP2	XJ17-5	MK759853	8
2017	China	NSP2	SCN17	MH078490	8
2017	China	NSP2	SCcd17	MG914067	8
2017	China	NSP2	GZgy17	MK144542	8
2017	China	NSP2	NADC30	MH500776.1	1
2017	China	NSP2	FJWQ16	KX758249	8
2017	China	NSP2	FJM4	KY412888	8
2017	China	NSP2	JX-CH	KY495780	8
2017	China	NSP2	SH-CH	KY495781	8
2018	China	NSP2	SCya18	MK144543	8
2018	China	NSP2	Henan-BY3	MN550954.1	8
2019	China	NSP2	Henan-JZ29	MN550957.1	8
2019	China	NSP2	SDRZ01	MZ322956	8
2020	China	NSP2	SD-QD-2101	MZ172971	8
2020	China	NSP2	SXSZ-2020	MW880772	8
2020	China	NSP2	JS2020	MZ342900	8
2020	China	NSP2	rJXA1-R	MT163314	8
2020	China	NSP2	SCcd2020	MW803134	8
2021	China	NSP2	BJ2021	OK095299	8
2021	China	NSP2	HY21	OL687155.1	8

**Table 2 vetsci-10-00325-t002:** The reference sequence of each PRRSV NSP2 strains in US.

Year	Aera	Sequence	Strain	Genbank Accession Number	Based on the ORF5 Lineage
1992	USA	NSP2	VR2332	EF536003.1	5
1993	USA	NSP2	ATCC VR2332	U87392.3	5
1996	USA	NSP2	SDSU73	JN654458.1	8
2003	USA	NSP2	JA142	AY424271.1	8
1998	USA	NSP2	RespPRRS MLV	AF066183.4	5
1999	USA	NSP2	Ingelvac ATP	DQ988080.1	8
1999	USA	NSP2	MLV RespPRRS/Repro	AF159149	8
2001	USA	NSP2	MN184A	DQ176019	1
2002	USA	NSP2	P129	AF494042.1	8
2004	USA	NSP2	NVSL 97-7895	AY545985.1	8
2006	USA	NSP2	Prime Pac	DQ779791.1	8
2008	USA	NSP2	NADC30	JN654459	1
2008	USA	NSP2	NADC31	JN660150	1

**Table 3 vetsci-10-00325-t003:** Recombination analysis of *Nsp2* gene.

Recombination Event	Recombinant Strains (Lineages)	Main Parental Strain (Lineages)	Minor Parental Strain (Lineages)	Recombinant Breakpoint	Recombination Analysis Method
1	CH-1R-2008 (8)	GD3-2005 (8)	CH-1a-1996 (8)	63~2935 (1884~1892)	RDP (P = 2.349 × 10^−149^) GENECONV (P = 8.169 × 10^−145^) BootScan (P = 1.298 × 10^−113^) MaxChi (P = 1.507 × 10^−46^) Chimaera (P = 3.911 × 10^−48^) SiScan (P = 3.619 × 10^−63^) 3seq (P = 3.334 × 10^−8^)
2	SD-YL1712-2017 (8)	BJ2021-2021 (8)	JSTZ1712-12-2017 (8)	7~2485 (784~823)	RDP(P = 1.915 × 10^−64^) GENECONV (P = 6.699 × 10^−96^) BootScan (P = 4.816 × 10^−98^) MaxChi (P = 5.325 × 10^−27^) Chimaera (P = 1.423 × 10^−25^) SiScan (P = 3.001 × 10^−31^) 3seq (P = 3.726 × 10^−9^)
3	MN184A-2001 (1)	NADC30-2008 (8)	QHD3-2017 (8)	63~2519 (1027~1061)	RDP (P = 5.700 × 10^−72^) GENECONV (P = 1.524 × 10^−45^) BootScan (P = 8.197 × 10^−71^) MaxChi (P = 3.496 × 10^−21^) Chimaera (P = 6.284×10^−22^) SiScan (P = 1.075 × 10^−29^) 3seq (P = 5.226 × 10^−55^)
4	NADC30-2017 (8)	SD-A19-2015 (8)	FJ1402-2014 (1)	772~889 (1414~1520)	RDP (1.308 × 10^−31^) GENECONV (P = 2.888 × 10^−27^) BootScan (P = 9.744 × 10^−32^) MaxChi (P = 4.598 × 10^−16^) Chimaera (P = 4.563 × 10^−16^) SiScan (P = 5.757 × 10^−19^) 3seq (P = 2.526 × 10^−43^)
5	SCya18-2018 (8)	SD17-36-2017 (8)	ZJXS1412-2014 (8)	17~2539 (352~482)	RDP (P = 1.874 × 10^−30^) GENECONV (P = 6.122 × 10^−8^) BootScan (P = 1.347 × 10^−29^) MaxChi (P = 7.742 × 10^−10^) Chimaera (P = 7.922 × 10^−12^) SiScan (P = 1.804 × 10^−10^) 3seq (P = 1.276 × 10^−10^)
6	Ingelvac-ATP-1999 (8)	CC-1-2006 (8)	HB-2(sh)-2001 (8)	33~2894 (907~1536)	RDP (P = 5.359 × 10^−10^) GENECONV(NS) BootScan (P = 7.970 × 10^−12^) MaxChi (P = 2.038 × 10^−14^) Chimaera (P = 2.298 × 10^−10^) SiScan (P = 5.881 × 10^−6^) 3seq (P = 1.257 × 10^−8^)
7	SCcd2020-2020 (8)	JS2020-2020 (8)	SCcd17-2017 (8)	762~903 (2112~2255)	RDP (P = 2.347 × 10^−4^) GENECONV(NS) BootScan (P = 3.109 × 10^−3^) MaxChi (P = 1.818 × 10^−7^) Chimaera(NS) SiScan (P = 4.693 × 10^−16^) 3seq (P = 2.094 × 10^−13^)

## Data Availability

All datasets are available in the main manuscript. The dataset supporting the conclusions of this article is included within the article.

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
