# Peer review of "Genetic Variability and Recombination of the NSP2 Gene of PRRSV-2 Strains in China from 1996 to 2021"

_vetsci, 2023, doi:10.3390/vetsci10050325_

Round 1

Reviewer 1 Report

The manuscript by Zhang et al. entitled “Genetic variability and recombination of the NSP2 gene of PRRSV-2 strains in China from 1996 to 2021” evaluated the NSP2 sequences from PRRSV-2 strains from China available through NCBI between 1996 and 2021. A total of 122 sequences were analyzed, and it was demonstrated that NADC-30-like and HP-PRSSV strains were the most prevalent in China during this time period (lineages 1 and 8, respectively). When evaluating the nucleotide and amino acid sequence identity, the authors determined a significant variation in sequence identity between strains (72.5%–99.8% and 63.9%–99.4% at the nucleotide and amino acid, respectively). Sequence variations are associated with deletions, insertions and substitutions, and recombination analysis demonstrated at least 5 recombination events with highest frequency among lineage 1. The authors conclude that such data will inform preventive and control measures. However, there are still many minor problems in the manuscript, which need further revision and improvement. The specific amendments are as follows:

1、 The clarity of Figures 6 and 7 needs to be improved.

2、 The punctuation marks in the article need to be carefully checked. For example, there is an extra ":" in line 12.

3、 In line 90, there is a more space.

4、 In line 114, “p<0.05” should be changed to “P<0.05”.

5、 In line 177, “path-ogenic” should be changed to “pathogenic”.

6、 Some terms in the article are incorrectly expressed. For example, “potent infectious disease” in line 22.

7、 The discussion needs to be more in-depth.

8、 The writing of the same concept should be consistent. For example, "Tables 1 and 2" in line 92 is inconsistent with "Figure 1 and Figure 2" in line 123.

Author Response

1、The clarity of Figures 6 and 7 needs to be improved.

Response: We have improved the clarity of Figures 6 and 7.

2、The punctuation marks in the article need to be carefully checked. For example, there is an extra ":" in line 12.

Response: We have deleted the ":" in line 11.

3、In line 90, there is a more space.

Response: We have deleted the space in line 96.

4、In line 114, “p<0.05” should be changed to “P<0.05”.

Response: We have changed “p<0.05” to “P<0.05” in line 120.

5、In line 177, “path-ogenic” should be changed to “pathogenic”.

Response: We have changed “path-ogenic” to “pathogenic” in line 170.

6、Some terms in the article are incorrectly expressed. For example, “potent infectious disease” in line 22.

Response: We have changed “potent infectious disease” to “serious infectious disease”, in line 21.

7、The discussion needs to be more in-depth.

Response: We have made overall revisions to the discussion, they are in lines 277-281 and 312-314.

8、The writing of the same concept should be consistent. For example, "Tables 1 and 2" in line 92 is inconsistent with "Figure 1 and Figure 2" in line 123.

Response: We have uniformly modified it to "Figures 1 and 2", in line 129.

Reviewer 2 Report

I reviewed the manuscript entitled “Genetic variability and recombination of the NSP2 gene of PRRSV-2 strains in China from 1996 to 2021” In this study authors performed an evolutionary analysis of PRRSV-2 using 122 sequences obtained from GenBank database, highlighting the relevance of recombination during the evolution of this virus in China.

Overall, I think this study lacks novelty. There is a previous study (Jiang et al., 2020)  doi.org/10.3389/fmicb.2020.00618 that covers the evolution of this virus in China from 1996 to 2017. The study conducted by Zhang et al. include 4 more years. Also, the study conducted by Jiang was conducted used full-length sequences. Instead, Jiang et al used just the NSP2 region. Why authors decided to exclude the other parts of the genome, considering the availability of this information at GenBank database?

Based on the above, I consider that the introduction of this study appears out of context. Multiple information is missing regarding the relevance of NSP2 (including breakpoints in NSP2) in the evolution of this virus in China. Based on the previously published information, update the introduction, and justify the conduction of this study.

Please, highlight the new findings of this study.  Based in the four-year window (2018-2021), What is different from the study published by Jiang et al., 2020?

Discussion should be drastically improved. It appears very repetitive with other section in the manuscript. Contrast the results obtained herein with previous publications.

Author Response

1、There is a previous study (Jiang et al., 2020)  doi.org/10.3389/fmicb.2020.00618 that covers the evolution of this virus in China from 1996 to 2017. The study conducted by Zhang et al. include 4 more years. Also, the study conducted by Jiang was conducted used full-length sequences. Instead, Jiang et al used just the NSP2 region. Why authors decided to exclude the other parts of the genome, considering the availability of this information at GenBank database?

Response: The conclusion about PRRSV in Jiang et al. 2020 article is based on the full-length sequence, which is an important perspective for analyzing the genetic diversity of PRRSV. However, due to the significant variation of the NSP2 gene in the PRRSV genome, research on the genetic variation of PRRSV mainly focuses on the highly variable NSP2 and ORF5 sequences. This study mainly uses different methods to study the genetic diversity of PRRSV based on the NSP2 gene, providing a new perspective for the genetic variation of PRRSV.

2、Based on the above, I consider that the introduction of this study appears out of context. Multiple information is missing regarding the relevance of NSP2 (including breakpoints in NSP2) in the evolution of this virus in China. Based on the previously published information, update the introduction, and justify the conduction of this study.

Response: We have made modifications to the introduction section by adding some introduction to NSP2 recombinant breakpoints and the correlation of NSP2 in evolution . They are in lines 69-71 and 76-78.

3、Please, highlight the new findings of this study. Based in the four-year window (2018-2021), What is different from the study published by Jiang et al., 2020?

Response: Jiang et al. divided PRRSV into 8 subgroups based on its full-length and analyzed the three types of Original HP-PRRSV, NADC30-like, and intermediate PRRSV, as well as deletion, insertion, and recombination. The final conclusion drawn is that recombination has played a very important role in virus evolution since 2006. Our research revolves around the PRRSV-2 NSP2 gene, dividing it into four lineages and conducting nucleotide and amino acid similarity, sequence alignment, and recombination analysis on NSP2. And we found that GM2-2011, QYYZ-2011, FJFS-2012 of lineage 3 have 37 amino acid insertions at positions 817-853 of NSP2. They are in lines 312-314 and 340-342.

4、Discussion should be drastically improved. It appears very repetitive with other section in the manuscript. Contrast the results obtained herein with previous publications.

Response: We have deleted the repetition and made overall revisions to the discussion, they are in lines 277-281 and 312-314.

Reviewer 3 Report

In this study, the authors compared the nucleotide and amino acid homologies of the NSP2 sequences of different PRRSV-2 lineages and examined phylogenetic relationships based on an analysis of the NSP2 sequences of 122 strains. They found that NADC-30-like strains, which are represented by lineage 1, and HP-PRRSV strains, which are represented by lineage 8, were the most prevalent in China from 1996 to 2021. What’s more, they identified deletions, insertions, and substitutions at multiple sites among the NSP2 sequences of PRRSV-2 strains. Recombination analysis revealed the occurrence of five recombinant events among the 135 selected PRRSV-2 strains, and that there is a high probability of recombination of lineage 1 strains. The findings enabled us to gain an in-depth understanding of the prevalence of PRRSV in China over the past 25 years.

Strength:

The results can provide a reference for the genetic evolution analysis of PRRSV.

Specific points:

1. As for “recombination analysis”, the authors pointed out “When potential recombinant events were detected based on RDP software (version 4.0), GENECONV, BootScan, MaxChi, Chimera, SiScan, and 3eq analyses, five or more methods were identified as genetic recombination and P < 0.05 in RDP software. The strains thus identified were considered recombinant strains. In addition, we used SimPlot(version 3.5.1) to confirm the detected recombination events.” But in the part of “3.5. Recombinant analysis”, only the results analyzed with RDP4.0 software and verified with SimPlot were described.

2. The interpretation of Figure 6 and Figure 7 is unclear, especially, the meaning of the ordinate is not explained clearly.

3. The number of strains used in phylogenetic analyses was inconsistent. It is 122 strains in line 120, but it is 129 strains in line 281.

4. Check the number of strains used in recombination analysis. The number of strains in Tables 1 and Table 2 adds up to 135 (line 236), but it is 133 in line 306.

Author Response

1、As for “recombination analysis”, the authors pointed out “When potential recombinant events were detected based on RDP software (version 4.0), GENECONV, BootScan, MaxChi, Chimera, SiScan, and 3eq analyses, five or more methods were identified as genetic recombination and P < 0.05 in RDP software. The strains thus identified were considered recombinant strains. In addition, we used SimPlot(version 3.5.1) to confirm the detected recombination events.” But in the part of “3.5. Recombinant analysis”, only the results analyzed with RDP4.0 software and verified with SimPlot were described.

Response: We have added a description of the RDP4 results. They are in lines 236-237.

2、The interpretation of Figure 6 and Figure 7 is unclear, especially, the meaning of the ordinate is not explained clearly.

Response: We have improved the clarity of Figures 6 and 7, and explained the meaning of the ordinate. They are in lines 255-257 and lines 266-267.

3、The number of strains used in phylogenetic analyses was inconsistent. It is 122 strains in line 120, but it is 129 strains in line 281.

Response: We identified 122 strains for phylogenetic analyses and updated the data in line 126.

4、Check the number of strains used in recombination analysis. The number of strains in Tables 1 and Table 2 adds up to 135 (line 236), but it is 133 in line 306.

Response: We identified 135 strains for recombination analysis and updated the data in line 320.

Reviewer 4 Report

Please find below my comments for the present manuscript instead.

Specific comments:

1- Abstract: The last sentence indicates that the work will contribute to "prevention and control" of PRRSV. This isn't clear, rather it informs our understanding of evolution and epidemiology.

2- Introduction, first paragraph: PRRSV-1 & -2 are no longer different genotypes, but different species.

3- Introduction, second paragraph: The nomenclature and classification of PRRSVs has changed, so please update this section, using a more recent reference.

4- Introduction, third paragraph: Please explain how monitoring of mutations will improve disease prevention and control.

5- Introduction, third paragraph: Please include reference(s) for the definitions of lineages or PRRSV-2.

6- Results, section 3.1: The last part of this section belongs more in the discussion than results.

7- Results, section 3.2: Please explain (in the materials and methods section) how and why the 15 strains were selected from each lineage, rather than using all of the 122 available originally.

Minor revision is suggested to address these points.

Author Response

1、Abstract: The last sentence indicates that the work will contribute to "prevention and control" of PRRSV. This isn't clear, rather it informs our understanding of evolution and epidemiology.

Response: We have modified "prevention and control" to "evolution and epidemiology", in line 42.

2、Introduction, first paragraph: PRRSV-1 & -2 are no longer different genotypes, but different species.

Response: We have made modifications to the basic information of PRRSV. They are in lines 54-55.

3、Introduction, second paragraph: The nomenclature and classification of PRRSVs has changed, so please update this section, using a more recent reference.

Response: We have made modifications to the basic information of PRRSV. They are in lines 64-68.

4、Introduction, third paragraph: Please explain how monitoring of mutations will improve disease prevention and control.

Response: We have modified "prevention and control" to "evolution and epidemiology", in line 87.

5、Introduction, third paragraph: Please include reference(s) for the definitions of lineages or PRRSV-2.

Response: We have added the definitions of PRRSV-2, they are in lines 79-80.

6、Results, section 3.1: The last part of this section belongs more in the discussion than results.

Response: We have moved this section to the discussion. They are in lines 286-294.

7、Results, section 3.2: Please explain (in the materials and methods section) how and why the 15 strains were selected from each lineage, rather than using all of the 122 available originally.

Response: Fifteen strains of PRRSV were selected for our study as they already contain the four lineages which we focus. We specifically chose representative classical strains from each lineage. We did not choose 122 strains because the NSP2 gene, which encodes 1016 amino acids, is relatively long. To focus on the differences of NSP2 genes in each lineage, we selected representative strains from four lineages for nucleotide sequence alignment. We believe that these 15 strains are sufficient to reflect the characteristics of the larger group of 122 strains to some extent.

Round 2

Reviewer 2 Report

I like to thank the authors for their responses, at this point, I don't have more concerns about this manuscript.